# Chimeric Peptides from *Californiconus californicus* and *Heterodontus francisci* with Antigen-Binding Capacity: A Conotoxin Scaffold to Create Non-Natural Antibodies (NoNaBodies)

**DOI:** 10.3390/toxins15040269

**Published:** 2023-04-04

**Authors:** Salvador Dueñas, Teresa Escalante, Jahaziel Gasperin-Bulbarela, Johanna Bernáldez-Sarabia, Karla Cervantes-Luévano, Samanta Jiménez, Noemí Sánchez-Campos, Olivia Cabanillas-Bernal, Blanca J. Valdovinos-Navarro, Angélica Álvarez-Lee, Marco A. De León-Nava, Alexei F. Licea-Navarro

**Affiliations:** 1Departamento de Innovación Biomédica, CICESE, Carretera Ensenada-Tijuana 3918, Ensenada C.P. 22860, Mexico; 2Instituto Clodomiro Picado, Facultad de Microbiología, Universidad de Costa Rica, San José 11501, Costa Rica

**Keywords:** NoNaBody, miniprotein design, VNAR and conotoxin fusion, chimeric peptide, protein scaffold

## Abstract

Research into various proteins capable of blocking metabolic pathways has improved the detection and treatment of multiple pathologies associated with the malfunction and overexpression of different metabolites. However, antigen-binding proteins have limitations. To overcome the disadvantages of the available antigen-binding proteins, the present investigation aims to provide chimeric antigen-binding peptides by binding a complementarity-determining region 3 (CDR3) of variable domains of new antigen receptors (VNARs) with a conotoxin. Six non-natural antibodies (NoNaBodies) were obtained from the complexes of conotoxin cal14.1a with six CDR3s from the VNARs of *Heterodontus francisci* and two NoNaBodies from the VNARs of other shark species. The peptides cal_P98Y vs. vascular endothelial growth factor 165 (VEGF_165_), cal_T10 vs. transforming growth factor beta (TGF-β), and cal_CV043 vs. carcinoembryonic antigen (CEA) showed in-silico and in vitro recognition capacity. Likewise, cal_P98Y and cal_CV043 demonstrated the capacity to neutralize the antigens for which they were designed.

## 1. Introduction

The use of antigen-binding proteins to block metabolic pathways has improved the detection and treatment of various pathologies associated with metabolite overexpression or malfunction. When compared with most commercial drugs, antigen-binding proteins have higher affinities for the binding sites of targets, which guarantees a greater specificity of treatment or detection. However, the use of antigen-binding proteins is not without problems, such as those related to their storage and transport, given that most of the available proteins are thermolabile. Another limitation of conventional antigen-binding proteins is their large size. In multicellular organisms, antigen-binding proteins are often unable to penetrate biological barriers and, thus, their treatment effects may be limited to the superficial layers of the tissues at the sites of their application. Various modifications have been made to antigen-binding proteins to reduce the disadvantages or limitations related to their use; however, most of these changes have tended to reduce the binding force.

Different strategies have been developed to generate proteins capable of neutralizing molecular targets of interest, such as variable domains of new antigen receptors (VNARs) and specifically those of cartilaginous fish antibodies (i.e., immunoglobulin new antigen receptors; IgNARs). Unlike conventional antibodies, IgNARs are heavy-chain homodimers that are linked by disulfide bonds and lack light chains. Each heavy chain contains a VNAR domain (~15 kDa) and five constant domains [1]. VNARs can be efficiently expressed in bacteria as active, soluble, and structured proteins. In addition, their small size and almost globular nature allow them to access gaps and hard-to-reach epitopes that antibody fragments cannot reach. Their high and rapid permeability also facilitates access to dense tissues [2].

As with conventional antibodies, the variety of IgNARs is based on the complementarity-determining region 3 (CDR3) of the VNAR domain, whose length can vary from 5 to 23 amino acid residues, although long loops of 15 to 25 amino acids are usually present, typically stabilized by disulfide bonds [3]. Nonetheless, there is a need for a small scaffold protein that can be used as a framework to generate chimeric proteins that retain their affinities for the binding site. Combining VNAR regions of antibodies from cartilaginous fish with scaffold proteins, such as macrocyclic peptides, may be a viable option to generate therapeutic and diagnostic molecules. Thus, macrocyclic peptides rich in disulfide bonds, such as cystine-stabilized alpha-helical peptides, constitute an emerging biomolecule class with potential therapeutic and diagnostic applications, as they share characteristics of both proteins (e.g., three-dimensional folding) and peptides (e.g., small size) [4,5].

Knottins are proteins of 20 to 50 amino acids in length. Despite their short size, knottins have a nucleus of antiparallel beta sheets stabilized by disulfide bonds instead of a “hydrophobic nucleus” [6,7]. These bonds play roles in the structure and function of peptides and confer the ability to form limited and well-defined three-dimensional structures. These well-defined structures can increase the potency, stability, selectivity, and permeability of knottins while lowering their susceptibility to degradation by proteases and allowing them to block peptides in active conformations. The mimicry and stabilization of secondary structures must be considered when designing inhibitors of protein–protein interactions for therapeutic targets [8]. For all the advantages of peptides rich in disulfide bonds, in this work, the use of cystine-stabilized alpha-helical peptides is proposed as a scaffold for protein engineering since they share several structural characteristics with knottins. Through rational design and protein engineering, mini-proteins constitute privileged scaffolds for drug development [6,9].

The current state of the art indicates no similar previous patented technologies or ideas like the one presented here. Patent number EP3277810 [10] contains information similar to the information in this study although with the opposite meaning. The recipient scaffold domain in patent EP3277810 is an antibody light-chain variable domain fused with a domain composed of a cysteine-rich peptide of 100 or fewer amino acids and, thus, the antibody acts as a carrier for the incoming peptide without contributing to target binding. Other studies have reported non-antibody peptides from other sources that were transplanted into antibody frameworks, such as binding naturally occurring integrin into the VH CDR3 region of a human antibody [11] and a peptide known to bind the thrombopoietin (TPO) receptor into several CDRs of an anti-tetanus toxoid antibody [12]. The present investigation aims to provide a conotoxin-derived scaffold protein capable of serving as a framework for generating chimeric antigen-binding proteins. Likewise, it provides a scaffold protein that, by inserting a CDR3 sequence from a parental VNAR, retains the recognition for the antigen presented by the parental VNAR.

## 2. Results

### 2.1. In-Silico Analysis (Homology Modeling, Molecular Dynamics, and Protein–Protein Docking)

#### 2.1.1. NoNaBodies Models

The chimeric peptides cal_P98Y, cal_T10, cal_CV043, cal_Tn16, cal_PK13, cal_SP240, cal_lis, and cal_AMA1 were modeled using a homology-based prediction program (Table 1). The 3D conformation structures of the molecules were refined and determined through molecular dynamics. To compare the structures, we used the reported model cal14.1a [13].

#### 2.1.2. Docking of VEGF_165_

Four possible interaction complexes between cal14.1a (as a negative control), parental VNAR (as a positive control), and the NoNaBody against a specific molecular target (e.g., cal14.1a, VNAR P98Y, and cal_P98Y against VEGF_165_) were generated for PPI analysis to identify the best binding complex of each tested molecule. In the VEGF_165_ analysis, the positive control VNAR P98Y [14] was identified with a total interaction score of −45.98 REU in the VNAR P98Y/VEGF_165_ complex (Figure 1a,b). The negative control complex cal14.1a/VEGF_165_ resulted in a total score of −14.88 REU (Figure 1c,d). When complexed with VEGF_165_, the NoNaBody cal_P98Y had a total score of −41.54 REU (Figure 1e,f). When identifying the interaction site with VEGF_165_, the CDR3 loop in the scaffold had the main interaction score in the complex, as it did in the positive control complex VNAR P98Y/VEGF_165_. The results of the interaction energy analysis of each model against VEGF_165_ are shown in Table 2.

#### 2.1.3. Docking of TGF-β

The interaction energy of VNAR T10, cal14.1a, and the NoNaBody cal_T10 were evaluated against TGF-β (Table 3). The positive control VNAR anti-TGF-β T10 resulted in a total interaction score of −27.24 REU, where the CDR3 of T10 was responsible for the primary interaction in the T10/TGF-β complex (Figure 2a,b). The negative control complex of cal14.1a/TGF-β resulted in a total score of −12.83 REU (Figure 2c,d). The NoNaBody cal_T10 against TGF-β had a total score of −34.55 REU (Figure 2,f). We identified the central interaction region with TGF-β to be the CDR3 loop in the scaffold for the cal_T10/TGF-β complex as it was in the positive control T10/TGF-β complex.

#### 2.1.4. Docking of CEA

In the anti-CEA analysis, the positive control VNAR CV043 resulted in a total interaction score of −18.76 REU. The CDR3 of VNAR CV043 had the higher score and the primary interaction in the VNAR CV043/CEA complex (Figure 3a,b). The negative control complex cal14.1a/CEA resulted in a total score of −8.53 REU (Figure 3c,d). The NoNaBody cal_CV043 in complex with CEA resulted in a total score of −22.16 REU (Figure 3e,f). We identified the interaction site with CEA. The CDR3 loop in the scaffold showed the main interaction score in the complex, as it did in the positive control complex of VNAR CV043/CEA. The results of the analysis of the interaction energy of each model against CEA are shown in Table 4.

#### 2.1.5. Docking of Other Pathological Targets

Designs of five specific NoNaBodies were evaluated for neutralization of the molecular targets for which they were created; cal_T16 for the tumor necrosis factor alpha (TNF-α), cal_PK13 for the protein convertase subtilisin/kexin type 9 (PCSK9), cal_SP240 for SARS-CoV-2 Delta SPIKE, cal_lis for lysozyme, and cal_AMA1 for the apical membrane antigen (AMA1) (Table 5).

### 2.2. In-Vitro Analysis

#### 2.2.1. Activity Evaluation of the Scaffold cal14.1a

By themselves, neither synthetic peptide favors the expression of TNF-⍺ in cells that are not exposed to pro-inflammatory stimuli (M0). However, in the presence of a pro-inflammatory stimulus, the cal141a peptide significantly decreases TNF-⍺ expression, while cal_P98Y does not modify the expression induced by LPS and IFN-y (Figure 4).

#### 2.2.2. VEGF_165_

##### VEGF_165_ Recognition Using ELISA

An ELISA was performed to evaluate the recognition of VEGF_165_ by the chimeric peptide cal_P98Y. Blocking with 3% bovine serum albumin (BSA) was used as a recognition control. The cal_P98Y peptide can recognize VEGF just like its parental VNAR. The scaffold conotoxin cal14.1a was used as a negative control, and its recognition was significantly lower compared to that of VNAR P98Y and the chimeric peptide (Figure 5).

##### Three-Dimensional In Vitro Angiogenesis Assay Based on Endothelial Cell Spheroids

Figure 6 shows the results of the in vitro angiogenesis assay. The anti-angiogenic effect of cal_P98Y was evaluated in endothelial cells (EC) with a spheroid-based three-dimensional test in triplicate. The spheroids were stimulated with VEGF_165_ and treated with cal_P98Y and VNAR V13_P98Y as a positive control. The cumulative sprout length (CSL) was quantified after 24 h of treatment. The control without VEGF_165_ stimuli (Figure 6a,f) showed low CSL and a statistically significant difference (*p* = 0.001) to the EC spheroids stimulated with VEGF_165_ (5 ng/mL; Figure 6b,f). The EC spheroids stimulated with VEGF_165_ and treated with cal_P98Y showed significant inhibition (*p* = 0.001) of sprout formation when compared to that of the EC spheroids that were only stimulated with VEGF_165_ (Figure 6b,e,f).

#### 2.2.3. TGF-β

##### TGF-β Recognition Using ELISA

An ELISA was conducted to evaluate the recognition capacity of the chimeric peptide cal_T10 of TGF-β as its parental VNAR. It was observed that cal_T10 can bind to TGF-β. The scaffold conotoxin cal14.1a was used as a negative control and showed no recognition capacity (Figure 7).

#### 2.2.4. CEA

##### CEA Recognition Using ELISA

An ELISA was performed to evaluate the recognition capacity of the chimeric peptide cal_CV043. Figure 8 shows that the chimeric peptide had greater recognition capacity than the parental VNAR CV043 when used as a positive control. The cal14.1a scaffold showed a significantly lower CEA binding capacity than the parent VNAR CV043 and the chimeric peptide cal_CV043.

##### CEA Labeling on the Surface of Cancer Cells

When evaluating the anti-CEA VNAR CV043 or the chimeric peptide cal_CV043 in the CEA-expressing HCT-116 cell line, an increase in mean fluorescence intensity (MFI) was observed (Figure 9, left). In contrast, the cancer cells MDA-MB-231 that did not express CEA showed no apparent change (Figure 9, right).

## 3. Discussion

The search for proteins and peptides to treat degenerative and infectious diseases has resulted in the search for various strategies to control the functioning and overexpression of proteins involved in altered metabolic pathways. Antigen-binding proteins as metabolic pathway-blocking molecules have improved the treatment of various pathologies associated with metabolite overexpression. The use of antibodies has been one of the most-applied methods by the pharmaceutical industry since they have a very high affinity for the binding site with the target antigen. However, using antigen-binding proteins is not without problems since most are thermolabile, making their storage and handling difficult. Another limitation is the large size of conventional antigen-binding proteins, which is why they can hardly cross the biological barriers of multicellular organisms, so their effect is limited to the superficial layers of the tissue where they are applied. For this reason, various modification strategies have been designed for antigen-binding proteins to improve their penetration rate or thermal stability. However, most modifications tend to reduce the binding force of the protein to the antigen-binding site. Due to the disadvantages described above, there is a need to provide antigen-binding proteins of reduced size that are thermostable and able to penetrate complex tissues.

One of these alternatives is VNAR antibodies, molecules with unique characteristics that allow them to bind to specific antigens [15,16]. This, coupled with their small size, chemical and thermal stability, and long CDR3s that allow for better access to biological targets, make them attractive molecules for therapeutic and diagnostic uses [15,17,18]. Several VNARs have been reported against different molecular targets, such as VEGF_165_ [19], TNFα [20], malaria [21], and SARS-CoV-2 [22]. Different methods have been employed to successfully obtain these VNARs, which include using immune libraries, semi synthetic libraries [23], and synthetic libraries [24]. Despite all of these advantages, VNARs have notable limitations. These limitations include the humanization of these molecules for their use as drugs in humans, the sizes of the molecules when they are used as drugs that require greater tissue penetration, and intellectual property barriers to their use in cancer treatments. Therefore, this study evaluated small chimeric peptides that were generated through the union of VNAR CDR3 with peptides rich in disulfide bonds (four times smaller than the VNAR) with the same interaction capacity as the parental VNAR.

Knottins are emerging molecules with potential therapeutic and diagnostic applications [4,5]. They are small peptides measuring 30 to 50 amino acids, with a core of antiparallel beta sheets stabilized by three disulfide bonds [6]. The alpha conotoxin cal14.1a is a cystine-stabilized alpha-helical peptide; however, it does not fall into the category of a knottin. Alpha conotoxin cal14.1a is a peptide that has 17 amino acids and two disulfide bonds between Cys3-Cys12 and Cys7-Cys17, which confer high structural stability [25]. Cal14.1a has not been found using NMR or X-ray crystallography, so homology modeling had to be carried out, and a structure for cal14.1a was obtained. This structure was compared with those obtained from the modeling results of the cal14.1a and cal14.1b conotoxins reported by Oroz et al. 2020. In both studies, the modeled peptides showed similar results to those of NMR-resolved conotoxins. Through in-silico and in vitro analysis, it has been shown that the peptide cal14.1a loses its activity as a conotoxin against its natural target when modified with the different CDR3s of VNARs previously reported. A relative expression test of TNFα was performed to evaluate whether the activity of the cal14.1a scaffold changed if its amino acid sequence was modified. We found that conotoxin cal14.1a reduced TNFα expression, while the chimeric peptide cal_P98Y did not (Figure 4). This result indicates that peptide cal14.1a is a privileged scaffold for rational protein design. By exchanging the native amino acid region for the CDR3 of a VNAR, cal14.1a loses its activity as a conotoxin [25].

Previously reported *Heterodontus francisci* (Horn shark) VNAR CDR3s were used against VEGF_165_, TGF-β, TNFα, PCSK-9, and SARS-CoV-2 SPIKE, including one CDR3 from *O. maculatus* (Carpet shark) VNAR and one CDR3 from *G. cirratum* (Nurse shark) VNAR. We demonstrated that binding with peptides rich in disulfide bonds linked to VNAR CDR3 works with any VNAR regardless of the shark species.

The strategy used in this study is based on in-silico tests used to model NoNaBodies through homology and molecular dynamics to later evaluate protein–protein interactions. The active loop of the conotoxin was removed and replaced with a CDR3 of a specific vNAR (Table 1). Eight examples of NoNaBodies were used in this study: cal_P98Y against VEGF_165_, cal_T10 against TGF-β, cal_CV043 against CEA, cal_Tn16 against TNFα (Appendix A), cal_PK13 against PCSK-9 (Appendix A), cal_SP34 against SARS-CoV-2 SPIKE (Appendix A), cal_lis against lysozyme (*G. cirratum*; Appendix A), and cal_AMA1 against AMA1 (*O. maculatus*; Appendix A). All of the modeling information of the NoNaBodies, parental VNARs, and the target molecules, as well as the docking analysis, can be found in the Appendix A. For practical purposes, this article covers only the first three examples (caL_P98Y against VEGF_165_, cal_T10 against TGF-β, and cal_CV043 against CEA).

The peptide cal_P98Y produced post-dynamic results, suggesting that subsequent docking analyses could be performed against VEGF_165_. In the docking analysis, the chimeric peptide obtained an interaction score of −41.54 REU, while its parental VNAR (VNAR P98Y) received a score of −45.98 REU. VNAR has already been tested in-silico, in vitro, and in vivo against VEGF [14], which makes it a good reference example for this NoNaBody creation system. On the other hand, the interaction region of the parental VNAR P98Y against VEGF is specifically within CDR3, as in the peptide cal_P98Y. This NoNaBody is related and binds in the same region of VEGF_165_ as the parental VNAR P98Y. To compare whether this interaction score has in-silico validity, VEGF_165_ was docked against the cal14.1a scaffold conotoxin. A total score of −14.88 REU was obtained. By comparing this result against that of the NoNaBody cal_P98Y, we can observe that it is much higher than that of the scaffold peptide, which confirms that our NoNaBody cal_P98Y does have a binding affinity for VEGF_165_.

After the docking analysis, the NoNaBody cal_P98Y was synthesized. Once the peptide was synthesized, a recognition ELISA against VEGF_165_ was performed (Figure 5), in which the parental VNAR P98Y had a reference absorbance. In contrast, the NoNaBody cal_P98Y had an absorbance value, which was significantly different to that of VNAR P98Y. The scaffold peptide cal14.1a was statistically significantly different than the VNAR P98Y and peptide cal_P98Y, which suggests that the chimeric peptide might have the ability to bind to VEGF_165_. Therefore, an analysis of the 3D spheroids of HUVECs was conducted to evaluate the neutralizing activity of cal_P98Y compared to that of the parental VNAR P98Y. cal_P98Y was found to neutralize VEGF_165_ almost in the same way as its parental VNAR (Figure 6), which indicates that the NoNaBody rationally designed against VEGF_165_ is a viable drug option as well as the P98Y VNARs.

To demonstrate that the scaffold peptide cal14.1a did not have neutralizing activity, it was tested as a negative control. We found that the neutralizing capacity of the scaffold against VEGF_165_ was not significant against the cal_P98Y and VNAR P98Y proteins.

The rationally designed peptide cal_T10 against TGF-β constitutes another example of a NoNaBody that is capable of successfully recognizing a cytosine for which it was designed. The NoNaBody cal_T10 and parental T10 VNAR were modeled following the same protocol as in the previous example. The docking analyses indicated that the chimeric peptide could have the same binding capacity as its parental VNAR since cal_T10 obtained a total score of −34.55 REU, while VNAR T10 obtained a total score of −27.24 REU. The negative control of the interaction model of cal14.1a against TGF-β received a low score (−12.83 REU) compared to that of the positive control and that of the chimeric peptide. After the in-silico analysis, the NoNaBody cal_T10 was synthesized, and a recognition ELISA was performed (Figure 7). We compared the binding capacities of the parental VNAR T10 as a positive control, the cal14.1a scaffold as a negative control, and the chimeric peptide cal_T10. The results showed that the cal_T10 peptide can bind with TGF-β, compared to the negative control cal14.1a with an unclear bind. This result supports the results obtained in-silico, which indicate that this rationally designed molecule is a potential TGF-β inhibitor.

The NoNaBody designed against CEA is another successful example of rational protein design using cal14.1a as a scaffold. The VNAR CV043 and the NoNaBody cal_CV043 anti-CEA were modeled using the strategy described in Section 4. After molecular dynamics modeling with quality structural parameters, the docking analysis proceeded. The VNAR CV043 vs. CEA achieved a total interaction score of −18.76 REU, while the chimeric peptide cal_CV043 obtained −22.16 REU. This result suggests that the cal_CV043 peptide has a higher binding capacity than its parental VNAR (CV043).

For this reason, the peptide was synthesized to conduct the necessary in vitro tests later to validate the in-silico results. In the ELISA assay, we could identify that the NoNaBody and the parental VNAR recognized the CEA antigen. To confirm the binding capacity of the cal_CV043 to CEA, a cytometry assay was performed to evaluate the expression of CEA in an HCT-116 cell line. As a negative control, the cell line MDA-MB-231, which does not express the CEA antigen, was evaluated. The results showed that the NoNaBody cal_CV043 binds to CEA expressed by the colon cancer cell line HCT-116, and the parental VNAR CV043 was obtained through a synthetic library using the CEA protein as antigen. On the other hand, in the negative control, the cell line MDA-MB-231 did not change when adding both anti-CEA proteins. These results suggest that the NoNaBody cal_CV043 has potential therapeutic use as a drug in treating some types of cancer where CEA is involved [26,27].

To validate the protocol for the rational in-silico design of NoNaBodies based on an alpha conotoxin and a CDR3 of VNAR from *H. francsisci,* several examples of NoNaBodies specific to other molecular targets, such as TNFα, PCSK-9, and SARS-CoV2 SPIKE, were considered. The different chimeric peptides showed promising results in terms of neutralizing the specific molecular targets for which they were designed when compared to the performance of the parental VNARs (see Appendix A). To demonstrate that the rational design system of chimeric peptides based on the binding of a conotoxin with a shark VNAR CDR3 also works with other VNARs from other shark species, two reported VNAR-based chimeric peptide designs were constructed. The first considered a shark anti-AMA1 VNAR from *O. maculatus* [28], and the second considered a shark anti-lysozyme VNAR from *G. cirratum* [29].

The NoNaBody cal_AMA1 resulted in a higher interaction score than its parental VNAR, and the scaffold peptide cal14.1a resulted in a lower total score than the NoNaBody. Therefore, the results suggest that the peptide cal_AMA1 may be a potential inhibitor of the AMA1 protein like its parental VNAR. The same scenario was apparent in the example of a NoNaBody based on the anti-lysozyme VNAR. This shows that the binding system of a VNAR CDR3 with an alpha conotoxin works with any VNAR regardless of the shark species, which opens the possibility of creating non-natural antibodies 1/40 of the size of an IgG that are able to recognize and neutralize like regular antibodies. These new proteins can be used as drugs when regular antibodies cannot penetrate tissues, such as for the treatment of diabetic retinopathy by neutralizing VEGF_165_.

The previously exposed construction and operation of NoNaBodies in-silico and in vitro show that these generated molecules have a binding strength of between 85 and 150% of the reference binding strength of the parental VNAR. The construction described above allows NoNaBodies to maintain the ability to bind to the same epitope as the parent antibody without losing effectiveness. In addition, due to the smaller size of the NoNaBodies obtained compared to the size of the parental VNAR, they have a greater potential capacity for tissue penetration. Similarly, due to their conformation, NoNaBodies present an improvement in thermal stability. However, an important issue for the generation of a NoNaBody is to maintain the solubility of the new protein; if no initial hydrophilic amino acids from the CDR3 are included, this could create a non-soluble NoNaBody with a lack of activity. We experienced this in the first attempt with the cal_P98Y NoNaBody.

## 4. Materials and Methods

### 4.1. In Silico Analysis

#### 4.1.1. NoNaBody Design

Each NoNaBody was designed for the neutralization of a specific molecular target. An in-silico and in-vitro tested parental VNAR with the ability to bind to or neutralize a specific antigen was used. Once the ability of the VNAR to bind to the molecular target for which it was expressed or synthesized was identified, an in-silico evaluation of it was performed. The region of the most important amino acids in the union with its antigen was evaluated. Once the binding region was identified, the design of a NoNaBody was carried out by changing the RAEK region (essential amino acids in the original function of alpha conotoxin) of the cal14.1a scaffold for the CDR3 region of VNAR with the greatest interaction of the parental VNAR/antigen complex.

#### 4.1.2. Homology Modeling

As explained in Section 3, after rational loop grafting conotoxin cal14.1a by inserting VNAR CDR3 into the scaffold, the three-dimensional structures of all chimeric peptides were predicted through homology modeling using MODELLER v. 9.19 [30] through a strategy known as “Advanced Modeling for Multiple Templates.” BLAST-P was used to identify the consensus template structures for the models. All chimeric peptides were modeled based on the three distinct protein scaffolds of the different conotoxins with more than 50% identity. The template PDB files were downloaded from the Protein Data Bank (PDB) and included PDB ID 1OMG [31], 1FEO [32], and 1MVI [33].

#### 4.1.3. Molecular Dynamics and Simulated Annealing

After homology modeling, the three-dimensional structures of the chimeric peptides cal_P98Y, cal_T10, cal_CV043, cal_Tn16, cal_PK13, cal_SP240, cal_lis, and cal_AMA1 were refined through simulated annealing calculations and parallelized with Nanoscale Molecular Dynamics 2.13 (NAMD) software [34], followed by an analysis and visualization of the results using the molecular graphics software Visual Molecular Dynamics (VMD; [35]) and PyMOL Molecular Graphics System v. 2.2.2 for Mac OS X. Next, force field simulations were conducted using the parameter obtained with Chemistry at HARvard Macromolecular Mechanics (CHARMM36; [36]) for all molecules in this study. Ramachandran plots of the molecules obtained with the PROCHECK server tool and ProQ-Protein quality prediction server were used for quality control purposes [37].

The simulations were performed in a box containing TIP3P water molecules as the solvent with periodic boundary conditions, where an NPT ensemble was assumed along with a constant number of particles (N) and constant isobaric (P) and isothermal (T) conditions. The pressure was set to 1 atm, and the temperature to 300 K. These conditions were iteratively coupled to the annealing and relaxation steps. During the annealing step, a temperature ramp from 0 to 805 K was used with an increase of 100 K at linear slope of 0.1 ns. Once annealed, the system was cooled to 300 K at a linear slope of 0.1 ns, to reach an equilibrium step of 1 ns. Each model of the NoNaBodies was annealed 1000 times, using the “steepest descendent method”. After annealing and cooling, all chimeric peptides were subjected to a molecular dynamics analysis at 300 K and 1 atm for 50 ns. The atom trajectory coordinates and energies were written to the disk every ten ps. The most thermodynamically stable protein conformation with the longest existence time was selected using VMD Clustering in the VMD software based on the RMSD values under 1 Angstrom. The relative population of the selected protein was always more than 50% of the 2500 models obtained after the molecular dynamics analysis. All MD analyses were conducted on the CICESE’s OMICA cluster, using 32 nodes with 96 processors for each one.

#### 4.1.4. Protein–Protein Docking

To predict the possible binding site of cal_P98Y to VEGF_165_, cal_T10 to TGF-β, cal_CV043 to CEA, cal_Tn16 to TNF-⍺, cal_PK13 to PCSK9, cal_SP240 to SARS-CoV-2 spike, cal_lis to a lysozyme, and cal_AMA1 to AMA1, we used the protein–protein docking protocol in the ClusPro web tool [38,39,40,41]. The docking algorithm evaluates different putative complexes with favorable surface complementarities. The resulting complexes were filtered, and we selected those with favorable electrostatic solvation and desolvation free energies for further clustering. The PyMOL Molecular Graphics System for Mac OS X was used to visualize and select the best results. The best binding affinities and orientation were used to select the possible VEGF_165_/cal_P98Y, TGF-β/cal_T10, CEA/cal_CV043, TNF-⍺/cal_Tn16, PCSK9/cal_PK13, SARS-CoV-2 SPIKE/cal_D240, Lysosyme/cal_lys, and AMA1/cal_AMA1 complexes.

The Peptidederive tool in the ROSIE server was used with its default settings to predict regions of protein–protein interactions (PPI). For the docking calculation, Peptidederive produced a plot with the possible PPI residues, which were ranked according to the Rosetta energy units (REU) of the Rosetta scoring function [42,43,44]. To improve the interpretation of our results, we used a parental VNAR created against VEGF_165_ with a synthetic and immune library as a positive control and reference for the REU values obtained in the in-silico analysis, which was confirmed through in vitro assays [14,24].

### 4.2. In Vitro Analysis

#### 4.2.1. Protein Synthesis

After in-silico analysis, the chimeric peptides anti-VEGF_165_ cal_P98Y, cal_T10 anti-TGF-β, and cal_CV043 anti-CEA were synthesized with adequate quality controls by Agentide Inc. (55 Madison Avenue, Suite 400, Morristown, NJ, USA).

#### 4.2.2. Activity Evaluation of the Scaffold cal14.1a

The effect of the activity of each synthetic toxin on the expression of the TNF-α gene was evaluated using the THP-1 cell line. A total of 2 × 10^5^ cells were exposed to 50 nM PMA (Sigma Aldrich, St. Louis, MO, USA) for 24 h at 37 °C with 5% CO_2_ for macrophage (M0) differentiation. Subsequently, the supernatant was removed, and a medium without PMA was added 24 h prior to each treatment to favor polarization to pro-inflammatory macrophages (M1). Cells were exposed to 100 ng/mL of LPS and 100 IU rh-IFN-y for 24 h. The effect of each toxin was evaluated in M0 and M1 cells using a final concentration of 5 µM of each recombinant peptide. Total RNA was extracted using the TRI (Sigma) reagent according to the specifications of the manufacturer, and 0.5 µg was reverse-transcribed using the superscript III kit (Invitrogen, Waltham, MA, USA). The expression of the TNF-α gene was evaluated for each treatment using β-actin as the reference gene.

#### 4.2.3. VEGF_165_

##### VEGF165 Recognition Using ELISA Assay

A total of 5 ng of VEGF_165_ was added to each well of an ELISA plate. The plate was incubated overnight at 4 °C. The remaining VEGF_165_ was removed by washing with PBS-Tween 0.05% (PBST). Then, 200 μL of blocking solution (3% BSA in PBS) was added to each well and incubated for 1 h at 37 °C. The blocking solution was removed, and the wells were washed three times with PBST. Later, 50 μL of cal_P98Y was added to each well in triplicate at 25 μg/mL and incubated for 1 h at 37 °C. BSA (1%) was used as a negative control, and VNAR P98Y (25 μg/mL) was used as a positive control. The solution was discarded, and the plate was washed three times with PBST. Then, mouse serum was added (polyclonal antibodies anti cal_14), and the plate was incubated for 1 h at 37 °C. Later, 50 μL of anti-mouse-HRP (diluted 1:1000 in 1% BSA-1X PBS) for cal_P98Y and 50 μL of anti-HA-HRP (diluted 1:1000 in 1% BSA-1X PBS) for VNAR V13_P98Y were added. The plate was incubated for 1 h at 37 °C. Then, the solution was removed, and the wells were washed five times with PBST. A total of 50 μL of TMB substrate was added, and the plate was incubated for 15 min at 37 °C. The reaction was stopped with 50 μL of 1 M HCL, and the absorbance was read at 450 nm.

##### Three-Dimensional In Vitro Angiogenesis Assay and Endothelial Cell Spheroid Model

In vitro angiogenesis was measured using a spheroid-based three-dimensional assay [45]. Human umbilical vein endothelial cells (HUVEC) were cultured in endothelial cell growth medium (ECGM; Cell Applications, Inc., San Diego, CA, USA) and incubated at 37 °C with 5% CO_2_. Only the cells cultured from passage three were used. HUVEC monolayers with 80% confluence were trypsinized and resuspended in ECGM supplemented with 10% fetal bovine serum (FBS) for cell counts. After calculating the dilution to generate spheroids of approximately 400 cells, the HUVECs were resuspended in ECGM supplemented with 10% FBS and 20% methocel at a dilution of 400 cells/100 μL. The cell suspension was distributed in a 96-well U-bottom non-adherent plate and incubated at 37 °C with 5% CO_2_ for 24 h. The spheroids were recovered from the plates in a 50 mL tube and centrifuged at 200× *g* for 10 min.

The previous solution was mixed with neutralized collagen solution (2.5 mg/mL) at 4 °C. The spheroid solution was quickly distributed in a non-adherent 24-well flat-bottom plate. A total of 1 mL of the solution containing approximately 40 spheroids was deposited in each well. The plate was incubated at 37 °C for 30 min to allow the collagen to jellify. The following treatments (100 μL equimolar) were then added: (1) basal medium (BM; control group), (2) VEGF_165_ (50 ng/mL) + BM, (3) VEGF_165_ (50 ng/mL) + P98Y VNAR (100 μg/mL), (4) VEGF_165_ (50 ng/mL) + cal14.1a (15 μg/mL), and (5) VEGF_165_ (50 ng/mL) + cal_P98Y (25 μg/mL). The plate was incubated for 4 h at 37 °C with 5% CO_2_. Furthermore, additional doses of VNAR P98Y and cal_P98Y anti-VEGF_165_ (in 50 μL of PBS) were added to the corresponding treatments, and the plate was incubated for 24 h. The spheroids were fixed by adding 1 mL of 3.7% formalin. Images of 15 spheroids from each treatment and experiment (duplicate tests) were captured using an inverted microscope (Olympus, Hamburg, Germany) and the digital image software Image-Pro. The in vitro angiogenesis was digitally quantified by measuring the cumulative length of the capillary-like sprouts that grew on each spheroid.

##### Statistical Analysis

An unpaired Student’s *t*-test was used to evaluate the differences between treatment groups. *p*-values < 0.05 were considered statistically significant.

#### 4.2.4. TGF-β

##### TGF-β Recognition Using ELISA Assay

In a 96-well plate, 250 ng of cal14.1a, cal_T10, and VNAR T10 were immobilized in a volume of 50 µL in triplicate and incubated for 12 h at 4 °C. The solution was discarded, and the wells were blocked with 300 µL of 3% BSA in 0.05% TBS-Tween and incubated for 12 h at 4 °C. Then, the blocking solution was discarded, and 50 µL of rhTGF-β1 at 5 µg/mL (Peprotech, Cranbury, NJ, USA) was added. After another incubation for 1.5 h at 37 °C, the solution was discarded, and three washes were performed with TBS-Tween 0.05%. A total of 50 µL of anti-TGF-β 1D11.16.8 antibody at 8.4 µg/mL (dilution 1:1000, BioXcell, Lebanon, NH, USA) was added, and the mixture was incubated for 1 h at 37 °C, after which the solution was discarded, and three washes were performed with TBS-Tween 0.05%. A total of 50 µL of m-IgGκ BP-HRP was added at a 1:3000 dilution (Santa Cruz Biotech, Dallas, TX, USA) followed by incubation for 1 h at 37 °C. The solution was discarded, and two washes with TBS-Tween 0.5% and three washes with TBS-Tween 0.05% were performed. Then, 50 µL of TBM reagent was added, and the mixture was incubated for 0.5 h at 37 °C. The reaction was stopped with 50 µL of 1 N HCl, and the absorbance was read at 450 nm. As a negative recognition control, the same procedure was followed but TGF-β was not added.

#### 4.2.5. CEA

##### CEA Recognition Using ELISA Assay

A total of 50 µL of CEA at µg/mL was added to in a 96-well plate and incubated for 12 h at 4 °C. The wells were blocked with 300 µL of 3% BSA in 0.05% TBS-Tween and incubated for 12 h at 4 °C. The blocking solution was discarded, and 1 µg each of cal14.1a, cal_CV043, and VNAR CV043 were immobilized in a volume of 50 µL in triplicate and incubated for 12 h at 4 °C. Then, the solution was discarded, and three washes were conducted with TBS-Tween 0.05%. A total of 50 µL of m-IgGκ BP-HRP was added at a 1:3000 dilution (Santa Cruz Biotech), and the mixture was incubated for 1 h at 37 °C, after which the solution was discarded, and two washes with TBS-Tween 0.5% and three washes with TBS-Tween 0.05% were performed. A total of 50 µL of TBM reagent was added, and the mixture was incubated for 0.5 h at 37 °C. The reaction was stopped with 50 µL of 1 N HCl, and the absorbance was read at 450 nm. As a negative recognition control, the same procedure was followed but CEA was not added.

##### CEA Labeling on the Surface of Cancer Cells

The CEA-expressing HCT-116 human colon carcinoma cell line (CCL-247, ATCC) was cultured in McCoy 5A media (Corning, New York, NY, USA) supplemented with 10% FBS (Corning) and 1% antibiotic antimycotic solution (Sigma-Aldrich, San Luis, MO, USA) in a humidified incubator (37 °C, 5% CO_2_). The breast cancer cell line MDA-MB-231 human (HTB-26, ATCC) was cultured in DMEM high-glucose media (Corning), supplemented and incubated under the same conditions as those of HCT-116 cells, and used as a negative control of CEA expression.

Colon and breast cancer cells were detached from the culture plate by incubating the cells for 10 min on ice, after which the monolayer was gently scraped using the plunger of an insulin syringe. The cell suspension was recovered from the plate and centrifuged at 180× *g* for 5 min at 4 °C. The supernatant was removed, and the cell button was resuspended in 1 mL of ice-cold PBS. The number of cells and their viability were determined using trypan blue, and the concentration of cells was adjusted to 5 × 10^6^ cells/mL. For each treatment, 100 μL of cells was centrifuged, resuspended in FACS buffer (PBS, 1% BSA, 2 mM EDTA, and 0.2% NaN_3_), and incubated for 10 min at 4 °C.

Subsequently, the following different treatments were added to the cell suspensions: (1) unstained cells, (2) cells + secondary antibody αHA-FITC, (3) cells + VNAR CV0-43 (5 μg) + αHA-FITC, (4) cells + cal-CV043 (5 μg) + cal14.1a serum + αmouse IgG1-AF647, and (5) cells + cal14.1a serum + αmouse IgG1-AF647. The primary antibody (CV0-43 or cal_CV043) was incubated for 1 h at 4 °C followed by a wash with PBS. Subsequent antibodies were incubated one by one for 15 min at 4 °C followed by a wash with PBS. The stained cells were analyzed using flow cytometry (Attune; Applied Biosystems, Waltham, MA, USA), and 50,000 total events were acquired. The cytometric analysis was performed with Flowjo software v. 10.4.

## 5. Patents

A patent has been submitted via PCT in Mexico with the number MX/a/2020/012914.

## Figures and Tables

**Figure 1 toxins-15-00269-f001:**
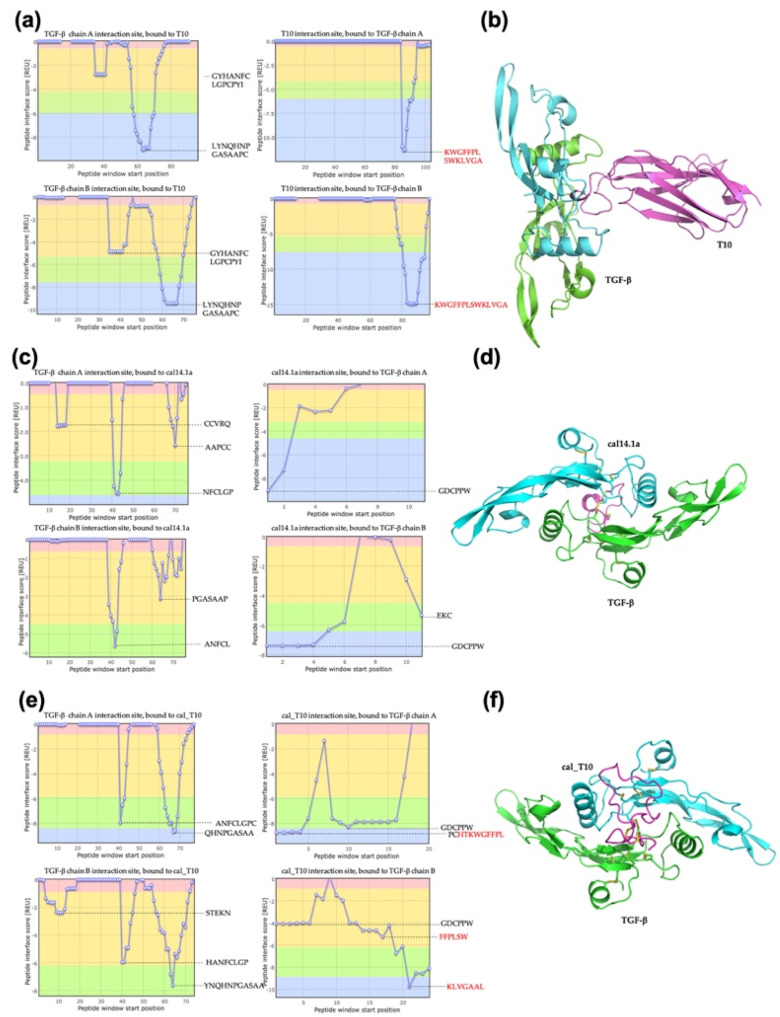
Protein–protein interaction from docking analysis in ClusPro and Peptiderive web tool of different molecules against VEGF_165_. (**a**) A plot of interaction regions of the positive control variable domains of new antigen receptor (VNAR) P98Y with VEGF_165_. In red are sequences of the complementarity-determining region 3 (CDR3) of VNAR. (**b**) VNAR P98Y/VEGF_165_ complex, P98Y (magenta cartoon), and VEGF_165_ (chain A green cartoon, and chain B cyan cartoon). (**c**) A plot of the regions of interaction of negative control cal14.1a and VEGF_165_. (**d**) cal14.1a/VEGF_165_ complex, cal14.1a (magenta cartoon), and VEGF_165_ (chain A green cartoon, and chain B cyan cartoon). (**e**) A plot of the interaction regions of chimeric peptide cal_P98Y with VEGF_165_. In red, CDR3 of VNAR P98Y added to the scaffold. (**f**) cal_P98Y/VEGF_165_ complex, cal_P98Y (magenta cartoon), and VEGF_165_ (chain A green cartoon, and chain B cyan cartoon).

**Figure 2 toxins-15-00269-f002:**
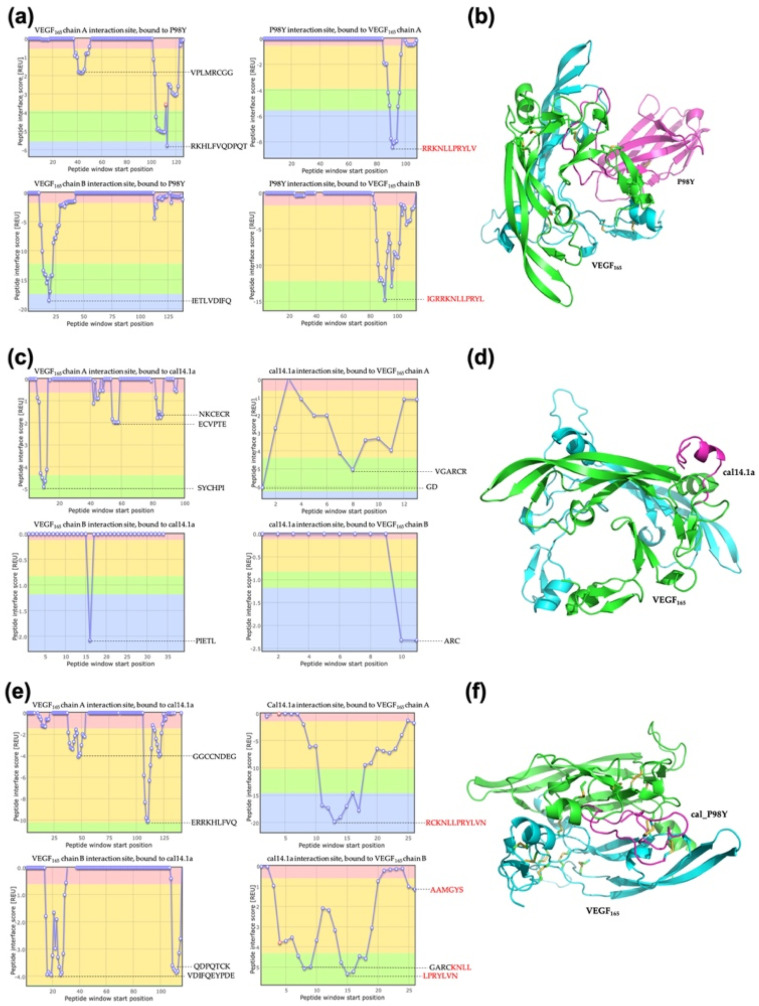
Protein–protein interaction from docking analysis in ClusPro and Peptiderive web tool of different molecules against TGF-β. (**a**) A plot of interaction regions of positive control VNAR T10 with TGF-β. The red text indicates the sequence of the complementarity-determining region 3 (CDR3) of the variable domains of the new antigen receptor (VNAR). (**b**) VNAR T10/TGF-β complex, T10 (magenta cartoon), and TGF-β (chain A green cartoon, and chain B cyan cartoon). (**c**) A plot of the interaction of amino acids of negative control cal14.1a against TGF-β. (**d**) cal14.1a/TGF-β complex, cal14.1a (magenta cartoon), and TGF-β (chain A green cartoon and chain B cyan cartoon). (**e**) A plot of interaction regions of chimeric peptide cal_T10 with TGF-β. CDR3 of VNAR T10 in the scaffold is shown in red. (**f**) cal_T10/TGF-β complex, cal_T10 (magenta cartoon), and TGF-β (chain A green cartoon, and chain B cyan cartoon).

**Figure 3 toxins-15-00269-f003:**
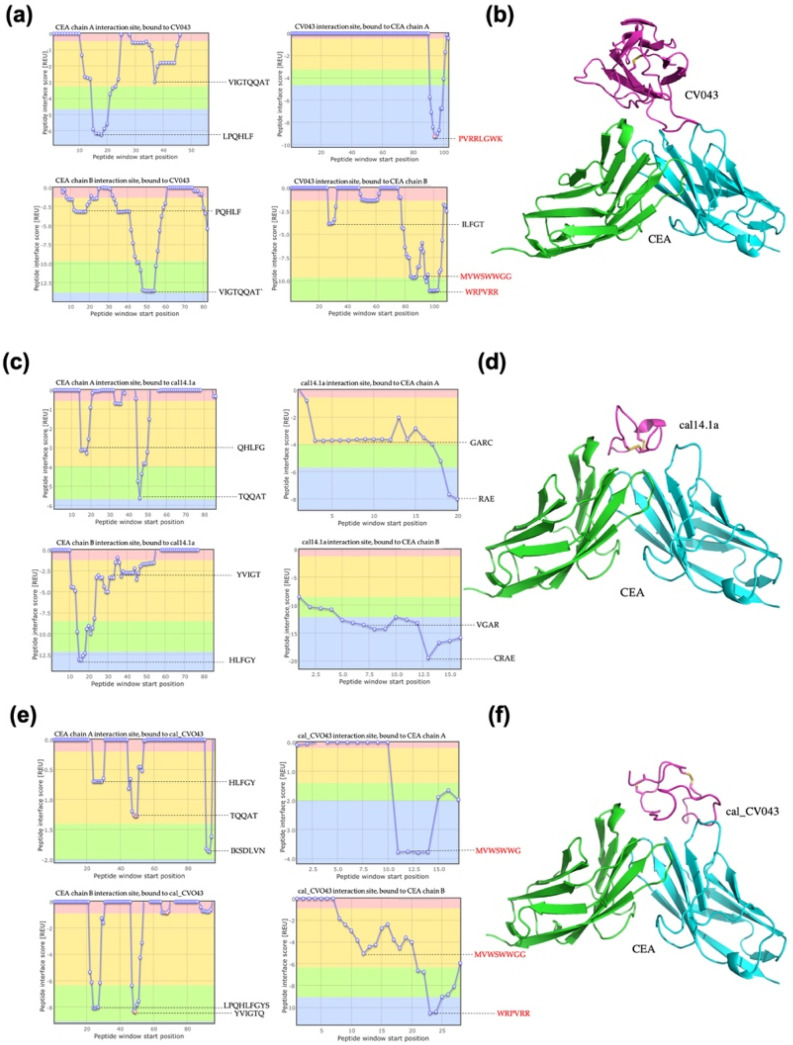
Protein–protein interaction from docking analysis in ClusPro and Peptiderive web tool of different molecules against CEA. (**a**) A plot of interaction regions of positive control variable domains of new antigen receptor (VNAR) CV043 with CEA, which shows the sequence of CDR3 of VNAR in red. (**b**) VNAR CV043/CEA complex, CV043 (magenta cartoon), and CEA (chain A green cartoon, and chain B cyan cartoon). (**c**) Plot of interaction of amino acids of the negative control cal14.1a against CEA. (**d**) cal14.1a/CEA complex, cal14.1a (magenta cartoon), and CEA (chain A green cartoon and chain B cyan cartoon). (**e**) A plot of interaction regions of the chimeric peptide cal_CV043 with CEA. The CDR3 of the VNAR CV043 in the scaffold appears in red. (**f**) cal_CV043/CEA complex, cal_CV043 (magenta cartoon), and CEA (chain A green cartoon and chain B cyan cartoon).

**Figure 4 toxins-15-00269-f004:**
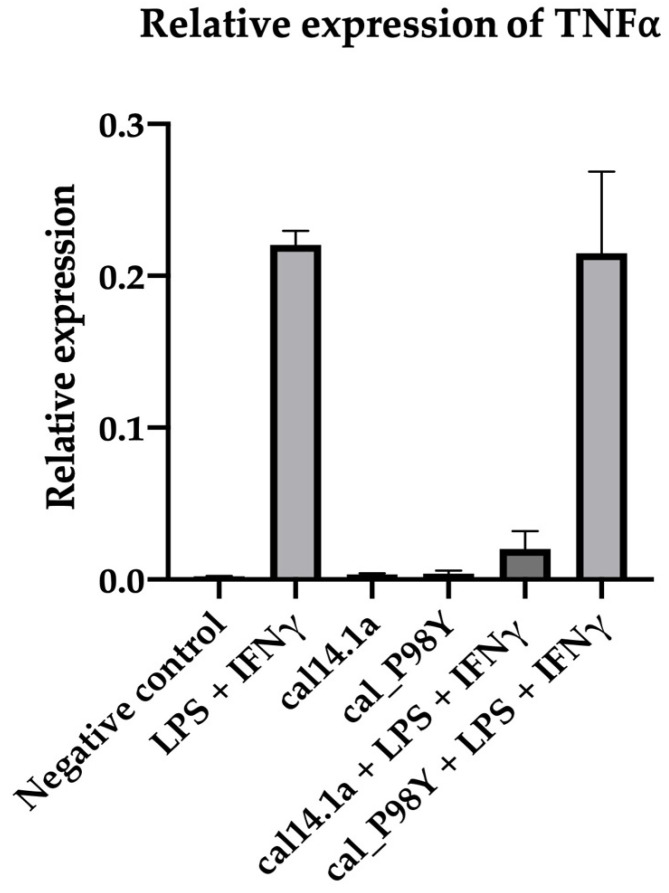
Relative expression of TNF-α. PBS used as a negative control.

**Figure 5 toxins-15-00269-f005:**
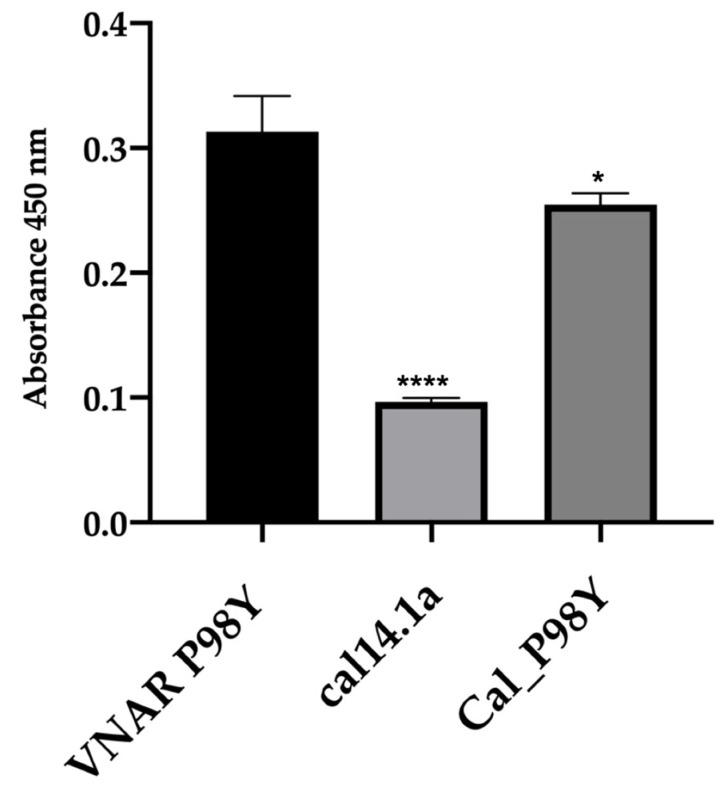
ELISA recognition assay of VEGF_165_ for cal14.1a (negative control), VNAR V13_P98Y (positive control), and chimeric peptide cal_P98Y. The difference in VEGF_165_ recognition of the VNAR P98Y compared to the cal14.1a peptide was significant (*p* ≤ 0.000 (****)). The NoNaBody cal_P98Y obtained a statistical difference of *p* ≤ 0.1 (*).

**Figure 6 toxins-15-00269-f006:**
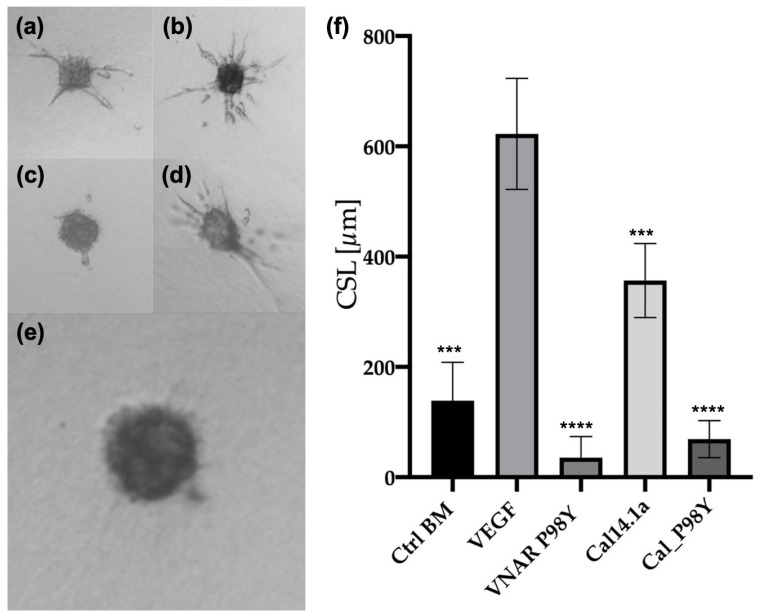
Three-dimensional in vitro angiogenesis assay based on collagen gel-embedded endothelial cell spheroids. Human umbilical vein endothelial cell (HUVEC) spheroids treated with VEGF_165_, anti-VEGF_165_ VNAR P98Y, and the chimeric protein cal_P98Y. (**a**) HUVEC spheroid without treatment (control basal medium (BM), *p* < 0.01 (***)). (**b**) Spheroid stimulated with VEGF_165_ (EC + VEGF_165_). (**c**) Positive control, spheroids stimulated with VEGF_165_ and treated with VNAR P98Y (EC + VEGF_165_ + VNAR P98Y). (**d**) Endothelial cell spheroids stimulated with VEGF_165_ and treated with cal14.1a (EC + VEGF_165_ + cal14.1a) *p* < 0.01 (***). (**e**) Endothelial cell spheroids stimulated with VEGF_165_ and treated with cal_P98Y (EC + VEGF_165_ + cal_P98Y). (**f**) A plot of cumulative sprout lengths (CSL) of capillary-like sprouts measured after 24 h of incubation. Angiogenesis sprouts were significantly (*p* < 0.001 (****)) inhibited when the spheroids were treated with VNAR P98Y and isolated chimeric protein cal_P98Y.

**Figure 7 toxins-15-00269-f007:**
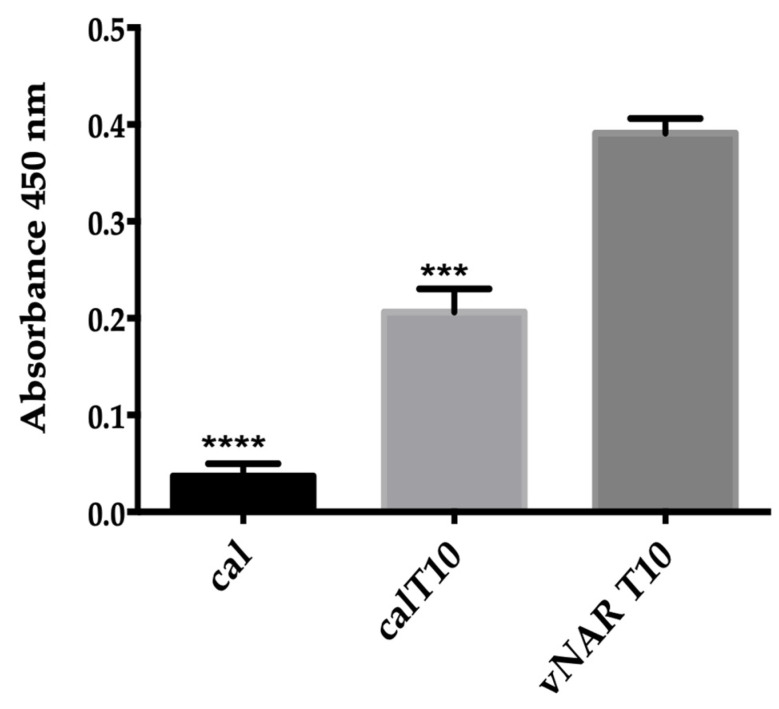
ELISA recognition assay of TGF-β for cal14.1a (negative control), VNAR T10 (positive control), and chimeric peptide cal_T10. The difference in TGF-β recognition of the VNAR T10 compared to the cal141a peptide was significant (*p* ≤ 0.001 (****)). The NoNaBody cal_T10 obtained a statistical difference of *p* ≤ 0.01 (***).

**Figure 8 toxins-15-00269-f008:**
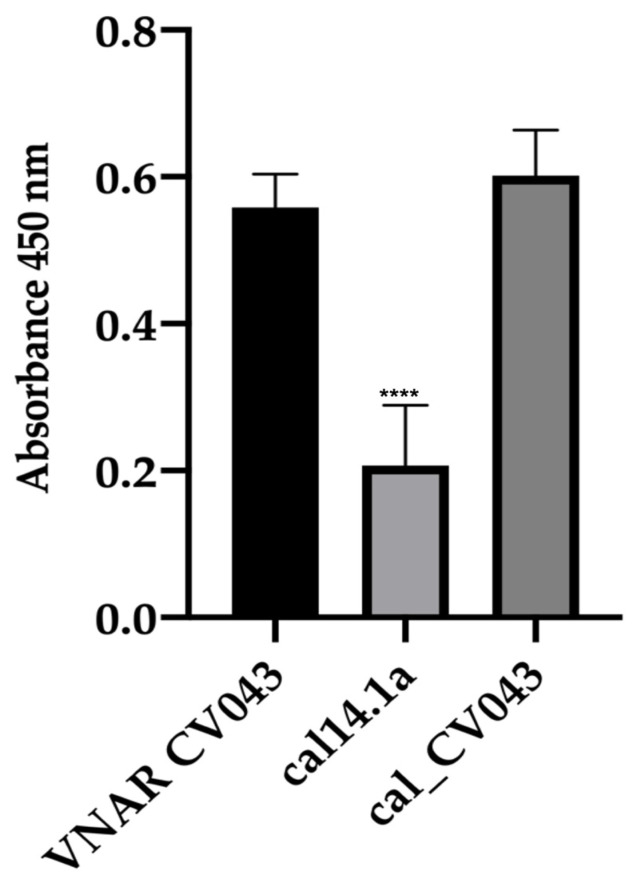
ELISA recognition of CEA for cal14.1a (negative control), VNAR CV043 (positive control), and chimeric peptide cal_CV043. The difference in CEA recognition of the VNAR CV043 compared to the cal141a peptide was significant (*p* ≤ 0.001 (****)). The NoNaBody cal_CV043 obtained no statistical difference.

**Figure 9 toxins-15-00269-f009:**
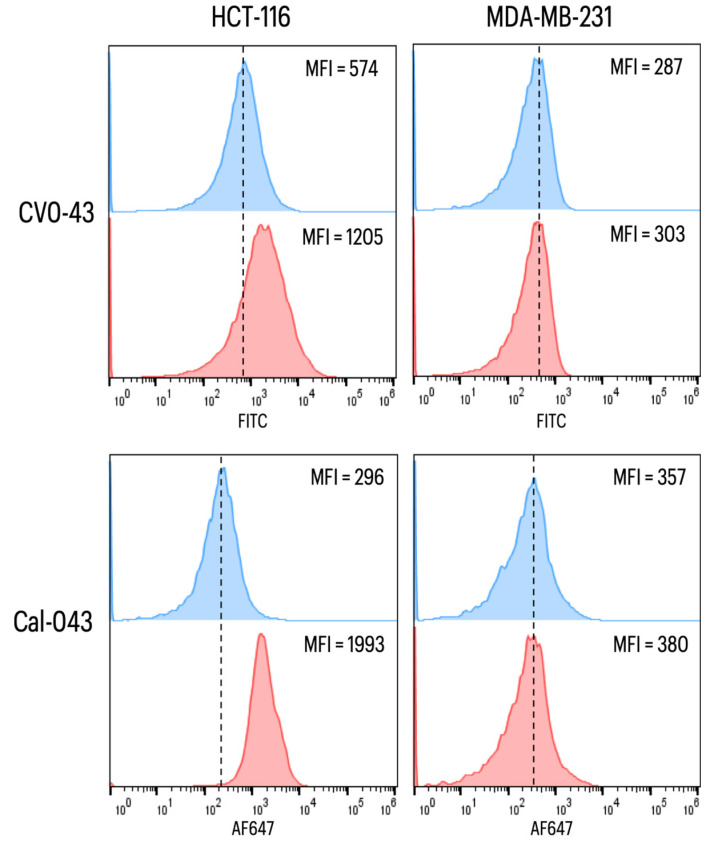
Representative histogram of anti-CEA in two cell lines and the MFI index. Colon cancer cells HCT-116 (CEA+) and breast cancer cells MDA-231 (CEA−) were stained with the anti-CEA antibodies (red plot), VNAR CV043, or the chimeric peptide cal_CV043, and the mean fluorescence intensity (MFI) was evaluated. The dotted line represents the mean of the negative control (blue plot), which were the only cells with secondary antibodies (αHA-FITC or αCal serum + αmouse IgG1-AF647).

**Table 1 toxins-15-00269-t001:** Chimeric peptide construction using a conotoxin from *Californiconus californicus* as a scaffold (in red) and the CDR3 of the VNAR (in black) of different shark species.

NoNaBody	Sequence
cal_P98Y	GDCPPWCGARCKNLLPRYLVNGIAAMGYSSSC
cal_T10	GDCPPWCGARCHTKWGFFPLSWKLVGAALINRSC
cal_CV043	GDCPPWCGARCDMVWSWWGGWRPVRRLGWKGWSC
cal_Tn16	GDCPPWCGARCKAQGLIDTSVRGLAVPGNCERCSSYHC
cal_PK13	GDCPPWCGARCARVWVSWVARAFFRGINFLPVFSC
cal_SP240	GDCPPWCGARCRAFGARARHEEGLEYYC
cal_lis	GDCPPWCGARCESRYGSYDAECAALNDC
cal_AMA1	GDCPPWCGARCFYSLPLRDYNYSLLC

**Table 2 toxins-15-00269-t002:** In-silico interaction complexes. The amino acids of complementarity-determining region 3 (CDR3) in contact with VEGF_165_ appear in red.

Molecule	Interaction Site Bound to VEGF_165_ Chain A	VEGF_165_ Chain A Interaction Site	Interaction Site Bound to VEGF_165_ Chain B	VEGF_165_ Chain B Interaction Site	Total Score (REU)
P98Y	RRKNLLPRYLV	RKHLFVQDPQT	IGRRKNLLPRYL	IETLVDIFQ	**−45.98**
cal14.1a	VGARCR	SYCHPI	ARC	PIETL	**−14.88**
cal_P98Y	RCKNLLPRYLVN	ERRKHLFVQ	KNLLPRYLVN	VDIFQEYPDE	**−41.54**

**Table 3 toxins-15-00269-t003:** In-silico interaction complexes. The amino acids of complementarity-determining region 3 (CDR3) in contact with TGF-β are colored in red.

Molecule	Interaction Site Bound to TGF-β Chain A	TGF-β Chain A Interaction Site	Interaction Site Bound to TGF-β Chain B	TGF-β Chain B Interaction Site	Total Score (REU)
T10	KWGFFPLSWKLV	QHNPGASAAP	KWGFFPLSWKLV	QHNPGASAAP	**−27.24**
cal14.1a	GDCPPW	ANFCL	GDCPPW	NFCLGP	**−12.83**
cal_T10	PCHTKWGFFPL	QHNPGASAA	KLVGAAL	YNQHNPGASA	**−34.55**

**Table 4 toxins-15-00269-t004:** In-silico interaction complexes. The amino acids of the complementarity-determining region 3 (CDR3) in contact with CEA are colored in red.

Molecule	Interaction Site Bound to CEA Chain A	CEA Chain A Interaction Site	Interaction Site Bound to CEA Chain B	CEA Chain B Interaction Site	Total Score (REU)
CV0-43	PVRRLGWK	LPQHLF	WRPVRR	VIGTQQAT	**−18.76**
Cal14.1a	RAE	TQQAT	CRAE	HLFGY	**−8.53**
Cal_CV043	MVWSWWG	IKSDLVN	WRPVRR	YVIGTQ	**−22.16**

**Table 5 toxins-15-00269-t005:** Comparison of the binding strength of the parental VNARs against their NoNaBodies.

Molecules		Binding Strength	Figure
TNF-⍺	NoNaBody cal_T16	−27.05 REU	Appendix A
	Conotoxin cal14.1a	−14.04 REU	
	VNAR Tn16	−33.01 REU	
PCSK9	NoNaBody cal_pk13	−28.20 REU	Appendix A
	Conotoxin cal14.1a	−8.98 REU	
	VNAR PK13	−20.39 REU	
SARS-CoV-2 Delta SPIKE	NoNaBody cal_SP240	−41.06 REU	Appendix A
	Conotoxin cal14.1a	−12.60 REU	
	VNAR SP240	−30.29 REU	
Lysozyme (*G. cirratum*)	NoNaBody cal_lis	−29.78 REU	Appendix A
	Conotoxin cal14.1a	−13.10 REU	
	VNAR A07	−32.29 REU	
AMA1 (*O.maculatus*)	NoNaBody cal_AMA1	−37.18 REU	Appendix A
	Conotoxin cal14.1a	−13.08 REU	
	VNAR 14I-1	−36.53 REU	

## Data Availability

All the information is at this document.

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
