# Peer review of "Chimeric Peptides from Californiconus californicus and Heterodontus francisci with Antigen-Binding Capacity: A Conotoxin Scaffold to Create Non-Natural Antibodies (NoNaBodies)"

_toxins, 2023, doi:10.3390/toxins15040269_

Round 1

Reviewer 1 Report

This manuscript “Chimeric peptides from Califonicosnus califonicus and Heterodontus fransisci with antigen binding capacity: a conotoxin scaffold to create a non-natural antibodies (NoNaBodies)” is very innovative work and I have not seen any similar protein/toxin engineering. I think it provides an excellent example of an application using a toxin (conotoxin) scaffold. The methods used in this study are a good combination of both in silico and in vitro approaches to validate protein binding. All my suggestions are quite minor.

Title:

There is an 'and' that should not be in italics and I would also suggest removing 'a' non-natural antibodies

Introduction:

Last paragraph of introduction is a bit colloquial in the writing, I would work to rephrase this paragraph.

There are two Figure 5, I believe one is meant to be labeled Figure 4.

Relative expression of TNFalpha (Figure 5) should have the negative control defined in the legend. 

Line 362: ‘with an unclear bind’

Line 363: in silico

Line 405: in silico and in vitro

Author Response

We appreciate all the comments of reviewer to our manuscript, which improve the quality of our work.

Title:

There is an 'and' that should not be in italics and I would also suggest removing 'a' non-natural antibodies

Answer: we agree with the suggestion, we add this comment to our manuscript

Introduction:

Last paragraph of introduction is a bit colloquial in the writing, I would work to rephrase this paragraph.

Answer: We did medications to the last paragraph.

There are two Figure 5, I believe one is meant to be labeled Figure 4.

Answer: Done.

Relative expression of TNFalpha (Figure 5) should have the negative control defined in the legend.

Answer: Done

Line 362: ‘with an unclear bind’ Answer: Done

Line 363: in silico.  Answer: Done

Line 405: in silico and in vitro. Answer: Done

Reviewer 2 Report

The submitted manuscript is a vital contribution to the field. The authors studied and are now reporting the design of chimeric peptides with target-driven recognition capacity for further development and clinical application.

For a final acceptance of the manuscript, improvements should be considered. Suggestions are as follows:

1-      Define what are the chimeric peptides cal_P98Y, cal_T10, cal_CV043, cal_Tn16, cal_PK13, cal_SP240, cal_lis, and cal_AMA1 the first time they appear in the text. It should be helpful for the reader's comprehension that the authors include a table showing the peptide constructions/sequences, even if the structures are schematically represented.

2-      Indicate the full names of target proteins that appear only as abbreviations in the text.

3-      If the used conopeptide template is not a knottin but a stabilized disulfide bond, it could be a cystine-stabilized alpha-helical peptide. It is recommendable to adjust the classification

4-      Include the common names of the species Califonicosnus califonicus, Heterodontus fransisci and other animal species the first time they appear in the text

5-      In the discussion, material and methods sections present the rationale of peptide chimeras synthesis and how the chimeric peptide peptides were synthesized, purified and characterized. Indicate some physicochemical properties.

6-      Discuss what kind of fusions for the design of chimeric peptide was considered, i.e., if the fusion was N- or C-terminal

7-      Indicate where the patent has been submitted with the number MX/a/2020/012914.

8-      The last statement in the discussion section, "The construction described above allows the NoNaBodies to maintain the ability to bind to the same epitope of the parent antibody without losing effectiveness. In addition, due to the smaller size of the NoNaBodies obtained, for the size of the parental VNAR, it has a greater capacity for tissue penetration. Similarly, due to their conformation, the NoNaBodies present an improvement in thermal stability" appears to extrapolate the experimental data. Were “tissue penetration” and “thermal stability” experimentally tested?

Author Response

We appreciate all the comments of reviewer to our manuscript, which improve the quality of our work.

  • Define what are the chimeric peptides cal_P98Y, cal_T10, cal_CV043, cal_Tn16, cal_PK13, cal_SP240, cal_lis, and cal_AMA1 the first time they appear in the text. It should be helpful for the reader's comprehension that the authors include a table showing the peptide constructions/sequences, even if the structures are schematically represented.

Answer: Done, we added a table with this information

  • Indicate the full names of target proteins that appear only as abbreviations in the text.

Answer: Done, lines 16 and 211-213

  • If the used conopeptide template is not a knottin but a stabilized disulfide bond, it could be a cystine-stabilized alpha-helical peptide. It is recommendable to adjust the classification.

Answer: Done, we have changed the term “cysteine knots”.

  • Include the common names of the species Califonicosnus califonicus, Heterodontus fransisci and other animal species the first time they appear in the text.

Answer: Done

  • In the discussion, material and methods sections present the rationale of peptide chimeras synthesis and how the chimeric peptide peptides were synthesized, purified and characterized. Indicate some physicochemical properties.

Answer: Done, line 470

  • Discuss what kind of fusions for the design of chimeric peptide was considered, i.e., if the fusion was N- or C-terminal.
  • Answer: Done, line 363

  • Indicate where the patent has been submitted with the number MX/a/2020/012914.

Answer: Done

  • The last statement in the discussion section, "The construction described above allows the NoNaBodies to maintain the ability to bind to the same epitope of the parent antibody without losing effectiveness. In addition, due to the smaller size of the NoNaBodies obtained, for the size of the parental VNAR, it has a greater capacity for tissue penetration. Similarly, due to their conformation, the NoNaBodies present an improvement in thermal stability" appears to extrapolate the experimental data. Were “tissue penetration” and “thermal stability” experimentally tested?

Answer: Done, the tissue penetration was not tested, we clarified this at line 468. We tested the thermal stability with non-representative modification.